# Institutional Change and Macroeconomic Variables in the ASEAN—Indonesia, Vietnam, and Cambodia: The Effects of a Trade War between China and USA

Fransiskus Xaverius Lara Aba 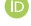

Faculty Economics and Business, Atma Jaya Catholic University of Indonesia, Jakarta 12930, Indonesia; fransiskus.lara@atmajaya.ac.id

**Abstract:** A trade war between the United States and China resulted in an increase in trade tariffs on imported goods entering each of these countries. Southeast Asian countries that have trade relations with the two countries, especially in terms of non-oil and gas exports of 25% to 35%, will be affected by export demand. Furthermore, the effects of the trade war will reduce gross domestic product (GDP) in Southeast Asian countries or the ASEAN and increase the current account deficit. On the other hand, the effects of the trade war that led to the decision of foreign investors to move their manufacturing base out of China will produce a flow of foreign investment that is ready to be captured by every ASEAN country.

**Keywords:** institutional change; macroeconomic variables; trade war; ASEAN countries





## 1. Introduction

The United States and China trade wars made each country impose an increase in trade tariffs so that the price of imported goods entering the country went up. In addition, US President Donald Trump urged US companies to move their operations out of China. The American president's call was based on an announcement from the Chinese government with a plan to impose additional import duties on incoming American products worth USD 75 billion (Nguyen and Canh 2020), which was later responded to by the American president by raising import tariffs of USD 550 billion for products imported from China (Garcia Herrero 2020).

The trade war is a term used by media and repeated by politicians and economists for United States (US) action who have raised import duties on imports of manufactured products from China and some other countries and received a backlash from the destination country. Action against China is taken because the US considers the government has jeopardized the interests of US nationals. China's export volume to the United States far exceeds the proportion of imports, so the US trade deficit with China since 2010, increasingly increased. To reduce the deficit trade, since June 2018 the US government has launched a strategy to increase import duties, especially towards China. In early May 2019, the US government increased tariffs on imports of Chinese commodities worth USD 200 billion from 10% to 25%.

China responded by imposing new import tariffs as of 1 June 2019, for items belonging to the US of USD 60 billion. China, too, is considering stopping purchasing the production of agriculture and Boeing aircraft from the US. This trade war entered a new chapter when Google from the US decided to connect their android system with Chinese mobile phone manufacturers, Huawei, to government prohibition. This has an impact significant for product users of Huawei. Hundreds of millions of Huawei phones can no longer access exclusive services of Google, such as Gmail, YouTube, Play Store, and Google Maps. US

mobile parts company, Lumentum, has also decided to stop sending spare parts to Huawei. The US believes Huawei is supported by the military, and so it is suspected of providing intelligence services for the Chinese government.

ASEAN countries have export trade relations with the United States and China, especially in terms of non-oil exports of 25% to 35%. With this trade war effect, it will affect export demand in the ASEAN. Products such as electronics, automotive raw materials, and plantation commodities have the most impact from falling export demand (Boltho 2020). By looking at the conditions that occur, the author will discuss institutional change and macroeconomic variables and the effects of the trade war in ASEAN countries. Expectations of the role of government in the cases of several ASEAN countries, such as Indonesia, Vietnam, and Cambodia, in addressing macroeconomic conditions lead to the importance of the institutional policies in order to support economic growth stability (Nguyen et al. 2018).

## 2. Literature Review

The most important policy and influence of macroeconomic variables in countries in Southeast Asia as a state institution is to maintain economic stability. This is measured from the current account balance of ASEAN countries that are still experiencing a deficit (Suk et al. 2017). The impact of the trade war seems to have two points of view, namely, reducing the level of exports and GDP growth, but on the other hand, produces an opportunity for the emergence of foreign investment flowing out of China, which will certainly have a positive impact on the ASEAN nation's economy.

### 2.1. Public Institutional Change

Public policies are made by the government to achieve certain goals in society, with the preparation through various stages (Adorisio et al. 2013; Battilana and D'aunno 2009; Greenwood et al. 2010; Greenwood and Suddaby 2014; Gábriel et al. 2014; Rowlinson et al. 2014). The institutional model is a traditional model in the policy-making process where the focus of this model lies in the organizational structure of government (Acs et al. 2017). Political activities are centered on government institutions, namely, the legislative, executive, and judiciary at the level of central (national), regional, and local government. *Public Policy Analysis* (Dunn 2015) is an applied social science discipline that uses a variety of research methods and arguments to generate and transform relevant information with the policies used in a specific political environment to solve policy issues related to the trade war due to the process. This trade war takes place on the worldwide stage, so it is hard to expect that it can be resolved bilaterally shortly; however, the world community has felt its impact. These impacts include, among others, the exchange rate of several countries that began to depreciate and the value of stocks (Robert and George 1999). The economic conditions of ASEAN countries Indonesia, Vietnam, and Cambodia, as well as the world, will be affected by the trade war, which will ultimately affect the domestic economy of countries in the world that have trade relations with China and the USA (Narjoko et al. 2009).

The stages of public policy making according to William Dunn are as follows: (i) Agenda setting is a very strategic phase and process in the reality of public policy. In this process, there is room to interpret what is called a public problem, and the public agenda needs to be taken into account (Moodysson and Zukauskaite 2014). If an issue has become a public problem, and gets priority on the public agenda, then the issue has the right to obtain more public resource allocation than other issues (Fratesi 2015). (ii) Policy formulation—problems that are already on the policy agenda are then discussed by policymakers. These problems are defined to find the best problem solving method (Healy 2016). The solution of the problem comes from various alternatives or available policy choices. Similar to the struggle with a problem to enter the policy agenda, in the policy formulation stage, each alternative competes to be selected as a policy taken to solve the problem. (iii) Policy legitimacy—the purpose of legitimacy is to authorize the basic processes of government. If

the act of legitimacy in a society is regulated by popular sovereignty, citizens will follow the direction of the government. However, citizens must believe that government action is lawful. Support for regimes tends to diffuse reserves of good attitudes and goodwill towards government actions that help members tolerate dissonance governance (Radosevic 2017). Legitimacy can be managed through the manipulation of certain symbols through which people learn to support the government. (iv) This means that policy evaluation is not only conducted at the final stage but is carried out throughout the entire policy process (Koschatzky et al. 2017). Thus, policy evaluation can include the stage of formulating policy problems, the proposed programs to resolve policy issues, implementation, and the stage of policy impact.

### 2.2. Macroeconomic Stabilization and Variable Policies

Stability policy in the macroeconomy is the duty of the government, and the presence of the government which functions as a regulator is expected to bridge all the needs and problems of society that arise in the practice of running the economy (Erjavec et al. 2015; North 1990; European Commision 2016). To conduct a stabilization policy, the government uses four policies, namely: (i) fiscal; (ii) monetary policy; (iii) wage and income policies; and (iv) industry and trade policy. Fiscal and monetary policies are policies that are seen from the demand side, while wage and income policies, as well as industry and trade policies, are policies that look at the supply side (Beverelli et al. 2018).

Fiscal policy is based on the theories of the British economist John Maynard Keynes, who states that an increase or decrease in income (tax) and the level of expenditure affect inflation, employment, and the flow of money through a country's economic system (Coleman 1987; Hałka 2015; Friedrich 1974). There are two main types of fiscal policy, namely: (i) Expansive fiscal policy, which is designed to stimulate the economy, and is most often used during times of recession, times of high unemployment, or other periods of low business cycles (Dell et al. 2018). This policy requires the government to spend more money, reduce taxes, or do both. The aim is to put more money in the hands of consumers, so they spend more and stimulate the economy. (ii) Contractionary fiscal policy, is used to slow down economic growth, such as when inflation grows too fast. In contrast to expansive fiscal policy, contractive fiscal policy increases taxes and cuts spending (Tran 2019).

Policymakers should ensure automatic corrections occur rather than engage in active fiscal and monetary policy on the concept of policy from theory. The new classical economists' policy suggestions are labor market reform, innovation through entrepreneurship, and ensuring the economy operates under a free market. These are all important for influencing aggregate supply. As a result, RBC theory rejects Keynesian economics or the view that in the short run economic output is primarily influenced by aggregate demand and monetarism, a school of thought that emphasizes the role of government in controlling the money supply (Kenneth 1997). Despite their rejection of the RBC theory, today's two schools of economic thought represent the foundation of mainstream macroeconomic policy.

The Austrian School believes that misguided government intervention causes business cycles. To boost economic growth, the government cut interest rates to artificially lower levels (Arshid 2017). Interest rates that are too low cause companies to accumulate too much equipment. That shifts aggregate demand to the right and creates an inflation gap in the economy.

NIE explores the notion of non-market institutions (property rights, contracts, revolutionary parties, etc.) to compensate for market failures. In the NIE approach, imperfect information, production externalities, and public goods are identified as the most important sources of market failure, thus necessitating the presence of non-market institutions (Kenneth 1988). On the other hand, in the neoclassical approach, the three variables above are assumed not to exist, so the transaction costs associated with these variables are considered non-existent. In addition, the NIE literature also adds several essential points about

institutional failures that are the cause of underdevelopment in many countries (Mark and Paul 2011).

The new political economy approach is defined as the relationship between political aspects, processes, and institutions with economic activities, such as production, investment, price creation, trade, consumption, etc. Based on this definition, it can be concluded that the political economy relates all political administration to economic activities carried out by the community and those introduced by the government. Economics and political science are different, but they can be compared, given that they have the same process (Charles and Wittkopf 2001). There are two main components of fiscal policy, which are as follows: (i) Taxation policy: The government gets income from direct and indirect taxes. Through its fiscal policy, the government aims to maintain as much progressive tax as possible. Furthermore, taxation decisions are very important for the economy for two reasons: (a) tax rates that are higher than usual will reduce people's purchasing power and will cause a decrease in investment and production and (b) a lower than the usual tax rate will make people spend their money, and this will cause inflation. (ii) Expense policy: This expenditure is carried out in the fields of development such as education, health, infrastructure, and others.

As for the following two points, the government budget is the essential instrument for government spending policies. The budget is also used to finance the deficit, meaning that government budgets are the most critical instruments that embody government spending policies (Rodríguez-Pose 2020). The budget is also used to finance the deficit to fill the gap between government spending and revenue: (i) Investment and policies: The optimal level of domestic and foreign investment is needed to maintain economic growth. In recent years, international capital flows or foreign direct investment (FDI) has increased dramatically and has become a tool to integrate the domestic economy with the global economy (Nguyen et al. 2014). (ii) Debt/surplus management: If the government receives more than what is spent, it is called a surplus. However, if the government spends more than income, then that is called a deficit. To finance the deficit, the government must borrow from domestic or foreign sources, or another option that can be taken is to print money to fund the deficit (Rodríguez-Pose and Garcilazo 2015).

Monetary policy includes policy measures implemented by the central bank to be able to change money supply or change existing interest rates, with the aim of influencing spending in the economy. The government determines four things that are the objectives of monetary policy, namely: (1) economic growth and income distribution; job opportunities; (2) price stability; (3) balance of payment balance. Meanwhile, monetary policy instruments include: (4) open market operations policy; (5) discount policy; (6) cash reserve policy; (7) strict credit policy; (8) moral encouragement policy.

Wages and income policies are government policies in terms of determining minimum wages, which are also very important, namely, in maintaining a balance between workers and employers. Equitable distribution of income does not mean that community income must be the same (Rodríguez-Pose and Ketterer 2020). Equity of opinion is undertaken so that the community's condition is getting better not lower. Inequality of income occurs because most of Indonesia's development is concentrated mostly in the big cities and there needs to be government attention supported by the community to jointly improve public quality services in the rural areas and suburbs. The application of taxes for high-income people must be more careful to cross-subsidize those whose economies are still low.

Industry and trade policy encompass foreign trade policy, which is a regulation made by the government that affects the structure or composition and direction of trade transactions and international payments. Because it is a part of macroeconomic policy, international trade policy cooperates well with fiscal policy and monetary policy. The objectives of foreign trade policy are as follows: (i) Protect national interests from negative influences originating from abroad such as the impact of foreign inflation on domestic inflation through imports or the effects of the world economic recession (global crisis) of Indonesia's export growth (Watson 2002); (ii) Protect national industries from competition

for imported goods; (iii) Maintain a balance of payments while ensuring adequate supply of foreign exchange, especially for the needs of imports and payment of installments and foreign debt interest; (iv) Maintain a high and stable rate of economic growth; (v) Increase employment opportunities.

*2.3. The Game Theory in Economy and Management*

This game theory aims to analyze the decision-making process of different competitors and involves two or more countries or company interests in running their business. The usefulness of a game theory is the methodology it provides for structuring and analyzing strategy selection problems. Using game theory, the first step is to explicitly determine the players' existing strategies and determine each player's preferences and reactions (Paul 2001). Two game strategies can be used in game theory: pure approach (each player uses a single system) and mixed strategy (each player uses a mixture of different methods). The refined design is used for games whose optimal results have a saddle point (a kind of balance point between the game values of the two players). Meanwhile, the mixed strategy is used to find the optimal solution for the game theory case that does not have a saddle point.

Several essential elements or concepts are crucial to solving each case with game theory. Here is a full explanation: (a) Number of players: Games are classified according to the number of interests or goals that exist in the game. In this case, it should be understood that the notion of "number of players" does not always mean the same as the "number of people" involved in the game. The number of players here means the number of groups or countries based on their respective interests or goals. Thus, countries or companies that have the same interests can be counted as a group of players. (b) Reward/payoff: Rewards/payoffs are the final results that occur at the end of the game. Regarding these rewards, games are classified into two categories, namely, zero-sum games and non-zero-sum games. A zero-sum game occurs when the sum of the tips of all players is zero, i.e., by taking each gain as a positive number and each loss as a negative number. Apart from that, it is a game of numbers—non-zero (Brooks 2010). In zero-sum games, every win for one player is a loss for the other player. The significance of the difference between these two categories of reward-based games is that zero-sum games are a closed system. At the same time, the non-zero-sum game is not like that. Most games are essentially zero-sum games. Various situations can be analyzed as a zero-sum game. (c) Game strategy in game theory is a strategy or a specific plan from a country or company as a player, as a reaction to actions that may be taken by players who are rivals. The games are classified according to the number of strategies available to each player (Chan 2014). If the first player has m possible strategies and the second player has n possible strategies, the game is called an m x n game. The importance of the different types of games based on the number of strategies is that the game is divided into finite games and infinite games. A little game occurs when each player's most significant number of strategies is limited or specific. In contrast, an endless game occurs when at least one player has an unlimited or indefinite number of strategies. (d) Game matrix: Every game analyzed with game theory can always be presented in the form of a game matrix. The game matrix is also called the reward matrix, which is a matrix in which all the elements are rewards from the players involved in the game. The rows represent the strategies of the first player, while the columns represent the other players' strategies (Brooks and Pallis 2013). Thus, the m × n strategy game is denoted by the m × n game matrix. Game theory assumes that the system available to each player can be calculated, and the rewards associated with them can be expressed in units, although not necessarily in monetary units. This is important for the game's completion, namely, determining the choice of strategy that each player will carry out, assuming that each player tries to maximize their minimum profit. The value of a game is the average reward/expected reward throughout the game, assuming both players always try to play their optimal strategy. Conventionally, the value of the game is seen from the side of the player whose strategies are represented by the rows of the reward matrix; in other words,

seen from the point of view of a particular player. A player is said to be fair if the value is zero, where no player gains an advantage or victory in an unfair game (unfair), and a player will win over another player, i.e., if the value of the game is not zero, in this case, the player's value is positive if the first player (row player) wins; otherwise, the game value is negative if another player (column player) wins (Charles 2017).

This is where the role of game theory is to determine which strategy will be decided by policymakers in a country to seize the market. The competition exemplified above can be identified to explain the concept of game theory which consists of several essential elements, namely: (1) The numbers in the payoff matrix, commonly called the game matrix, show the results (payoffs) of different game strategies; these results are expressed in the form of practical measures such as money, percentage market share, or utility. (2) The maximizing player is the player who is in the row and who wins/gains the game, while the minimizing player is the player who is in the column and suffers a loss. (3) Game strategy is a series of activities or a comprehensive plan of a player as a reaction to the behavior of their competitors. In this case, the strategy or program cannot be undermined by other competitors. (4) The rules of the game are the patterns in which the players choose a strategy. (5) The value of the game is the payoff result estimated by the player throughout the series of games in which each player uses their best strategy. The game is said to be fair if the value of the game is equal to zero and vice versa. (6) Dominance is a condition where the player with each payoff in the strategy is superior to each related payoff in an alternative method. The dominant rule is used to reduce the payoff matrix size and computation effort. (7) Optimal strategy is a condition wherein a series of game activities a player is in the most advantageous position regardless of the needs of their competitors. (8) The purpose of the model is to identify the optimal strategy or plan for each player (Bureau of Economic Analysis 2007).

### 3. Methodology

According to World Bank data, as of the period June–August 2019, 33 registered companies in China have announced plans to move or expand their manufacturing base out of China (World Bank 2019b). Twenty-three of these companies decided to enter Vietnam and ten other companies decided to enter countries such as Cambodia, India, Malaysia, Mexico, Serbia, and Thailand. This fact confirms the lack of potential of several ASEAN countries such as Indonesia, Singapore, and the Philippines in the eyes of foreign investors (Coxhead et al. 2019). The opportunities created by the trade war between the United States and China apparently did not necessarily have a positive impact on ASEAN countries because the shifting of the manufacturing base of American companies from China could not be captured by Indonesia, Singapore, and the Philippines.

The following are the factors that hinder countries in the world, especially developing countries, in capturing investment opportunities from outside: Internal procedures are inefficient in terms of cost and time. When the manufacturing base has been built, then investors will think about the life cycle prospects of the industry so that it can develop and so that it ultimately contributes to an increase in gross domestic product (Suddaby et al. 2014; Rutherford 1994).

Therefore, the essence and ease of export and import need to be increased, considering that domestic output requires raw materials from import activities and will be traded internationally through export activities. According to statements from the World Bank, several Southeast Asian, Middle Eastern, and Eastern European countries as well as Africa have not found their position in the global supply chain in terms of exports and imports due to the following factors (World Bank 2019c): (i) Inefficient import procedures such as the imposition of pre-shipment inspection by two parties, namely, surveyor companies located in each of these countries, import approval requiring recommendations from the Ministry of Industry that uses discretion in determining the types of production inputs and how much can be imported, and additional standard adjustments from third parties to the national standards in each country. (ii) There are import tariffs applied for production

inputs. (iii) Lack of adequate human resources for production engineers, process engineers, design engineers, production planning and inventory managers, and human resources managers. (iv) High logistics and resource costs compared to other neighboring countries. (v) From making policies that are not by practice, such as the exemption from import procedures for production goods in an area of a free-trade zone, that turns out to be incompatible with practice (Thornton 2001).

There is a recommendation for approval of imports to be a maximum of five days but this is reaching 3–6 months. There is authority uncertainty in determining which imports require a recommendation letter or not, so that some industries must stop their production lines. Furthermore, there is inefficiency in the policy in the case of each importer obtaining a written statement from the producers who cannot channel inputs according to specifications (Wang et al. 2019).

## 4. Results and Discussion

### 4.1. Trade War and ASEAN Macroeconomic Variable Change

ASEAN countries that adopt an open economic system will have exposure to the influence of macro conditions from abroad such as the influence of a trade war between the United States and China. When a trade war occurs, each country imposes an increase in trade tariffs so that the price of imported goods goes up.

The rising price of imported goods will reduce the volume of trade so that the exporting country will reduce its production activities, which will impact on the decline in the level of exports. When the level of exports decreases, commodity exporting countries to the United States and China will stop exporting because demand for the commodity stops due to stopped production activities. ASEAN, as one of the commodity exporting countries to China and America, needs to get around the impact of the trade war by considering economic policies that involve macroeconomic variables.

Regarding the limited company in the Ministry of Law and Human Rights of the Republic of Indonesia, in Article 8 paragraph (2) letter a, it is necessary to clarify the nationality of the founders to establish a company. An Indonesian legal entity in a company is established by an Indonesian citizen or an Indonesian legal entity. However, foreign nationals or foreign legal entities are allowed to establish an Indonesian legal entity in the form of a company as long as the law governing the company's line of business allows, or the establishment of the company is regulated by a separate law. If the founder is a foreign legal entity, the number and date of legalization of the founding legal entity is a document similar to that, including a certificate of incorporation. If the founder is a state or regional legal entity, a government regulation concerning participation in the company or a provincial law concerning regional involvement in the company is required.

Examples of cases when foreign investors open business opportunities in Indonesia, show that the ease of licensing needs to be at the beginning to build positive sentiment from investors. The following is an example of the case of the estimated time needed to build a company in Indonesia based on Law Number 40 of 2007 (Table 1).

Examples of cases of foreign investors who want to establish a company in Indonesia should turn to the Investment Coordinating Board appointed as the foreign investment service's agency. Strict procedures for foreign investors who want to build a company in Indonesia are accommodated through the management services provided by local companies that have a specialty in completing all procedures required by the Investment Coordinating Board. However, this process costs up to USD 3000 and takes about 10 weeks to process.

**Table 1.** Example of the case of the estimated time needed to build a company in Indonesia.

| | Estimated Time(Days) |
|---|---|
| Principle License and Business License | 7 |
| Deed of Establishment (containing the Articles of Association) legalized by a public notary | 1 to 2 |
| Legalization of the legal entity status of the FDA by the Ministry of Law and Human Rights | 10 |
| Domicile letter from the local district authority | 3 |
| Tax Identification Number and taxable entrepreneur confirmation from the tax office | 3 |
| Company Registration Certificate from the agency for integrated licensing services | 14 |
| Manpower Report and Company Welfare Report from the sub-department of the Ministry of Manpower | 7 |

Source: Ministry of Law and Human Rights of the Republic of Indonesia 2007.

Based on the macroeconomic concept, GDP will be affected by several economic variables according to the formulation of when the level of exports (Malesky et al. 2015) of ASEAN countries' commodities have declined due to trade war, which will reduce the value of Y (GDP). Declining levels of demand for commodities from the United States and China cause commodities to become ineffective so that commodity prices fall. This can be seen through the following shifts in IS (Figure 1).

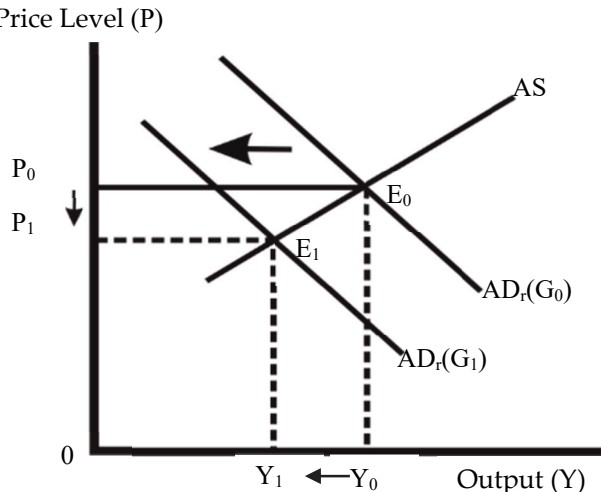

**Figure 1.** Shifts in income and interest rates (IS) due to changes in fiscal policy.

In the case of Indonesia, according to projections from the World Bank, the impact of trade war, which caused the global economic downturn, has the effect of weakening Indonesia's economic growth going forward (Warjiyo 2017). The decline in GDP is expected to fall to 4.6 in 2022.

The World Bank states that the direction of Indonesia's economic growth will continue to decline to the level of 4.9% in 2020, due to (1) low productivity; (2) the slowdown in the growth of the workforce; (3) a decrease in commodity prices; and (4) continuation of the US and China trade war. China's economy fell by 1%, then Indonesia's GDP can fall by 0.3%. In the case of the 2009 global recession, global economic growth was down by 6.2% from 2007, commodity prices fell, and the Indonesian economy slowed by 1.7%. It is believed that monetary and fiscal conditions in Indonesia will be limited in the future. Projecting the opportunity for a 7DRR decrease is relatively limited, which is only a 25% to 5% level in 1H20F. In assessing that, the positive effects of lowering interest rates are fading, and

even aggressive declines can trigger capital outflows in the future. Fiscal stimulus from the plan to reduce corporate tax from 25% to 20% is estimated to be effectively implemented in 2021. Therefore, lowering the projected corporate earnings growth from 8–10% in FY19E to 5.28–7.98% in FY19E FY20F. In my view, investment risk in Indonesia still lies in the: (1) the current account deficit (CAD), which is still widening; (2) the weakening of the exchange rate, which is still estimated at the level of 15,000 in FY20F; and (3) capital outflow. Every year, Indonesia at least still needs external funding of around USD 16 billion to finance the CAD, where this value will increase in an outflow. On the other hand, the low level of labor productivity and the inefficiency of the bureaucracy in Indonesia have pushed down the contribution of FDI to Indonesia's GDP since 2014. The positive impact of the trade war is more beneficial for Vietnam, Thailand, Malaysia, Singapore, and Taiwan. Twenty-three Chinese companies that went public have announced plans for Vietnam expansion, with the remaining ten to Cambodia, India, Malaysia, Mexico, Serbia, and Thailand in 2019. Researchers believe that FDI growth is one of the essential indicators to increase investor confidence shares amid the very lack of catalysts for the Indonesian stock market at this time (Figure 2).

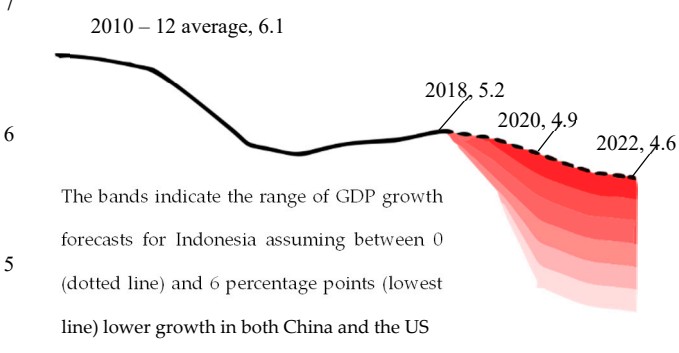

**Figure 2.** Indonesia's economic growth 2010–2022. Source: World Bank 2020.

The impact of the decline in GDP in the future will further depress the condition of the Indonesian economy and shift IS to the left so that commodity prices will further decline. This might happen considering that Indonesia's largest export destination countries in the aggregate are China and America. When the level of exports declines, Indonesia's current account balance will shift towards a deficit. The following (Figure 3) is Indonesia's current account balance during the trade war period.

For example, the current account deficit in Indonesia experiences a deficit when the export level is lower than the import level. When the level of exports is lower, the flow of money into Indonesia will be reduced so that the potential for capital outflows is more frequent, considering that Indonesia is also an importing country. The imbalance between imports and exports also means the rupiah has been weakened since March 2018, when the US began to impose trade tariffs for the first time on Chinese products according to the following graph (Figure 4).

Current Account Balance (Million USD)

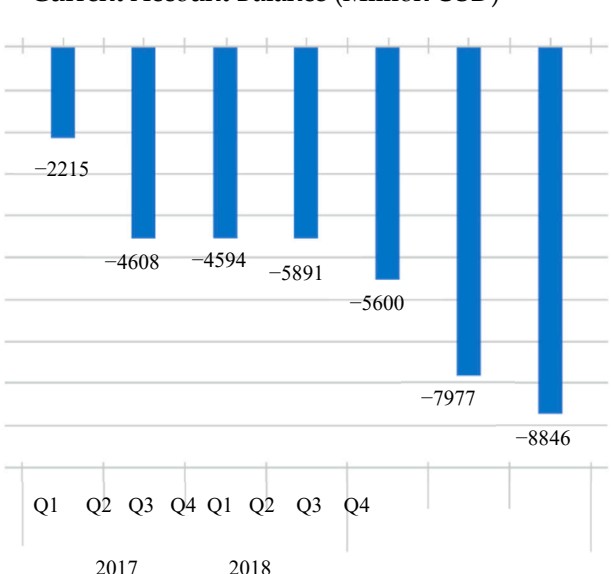

**Figure 3.** Effectiveness of monetary policy. Source: Report of the Central Bank of Indonesia and the Central Statistics Agency 2019.

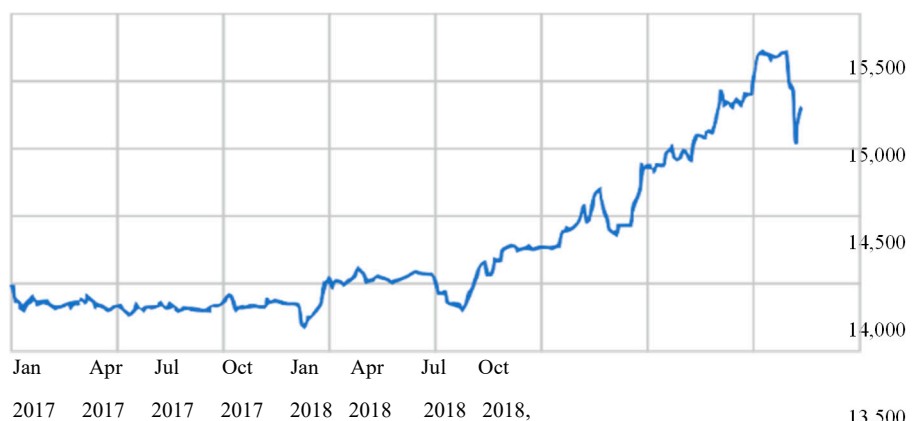

**Figure 4.** Indonesian imports and exports in US dollars. Source: National Central Agency Report 2019.

As a result of the depreciation of the rupiah, the Central Bank of Indonesia finally adopted a monetary contraction policy by raising the benchmark interest rate. This policy is reflected through the LM Figure 2 shifting to the following (Figure 5).

The policy of raising the benchmark interest rate aims to strengthen the attractiveness of Indonesian domestic financial assets in the eyes of investors so that investors hope to remain interested in investing in Indonesia and capital outflows can be minimized. In the end, maintained investment flows will increase the exchange rate against the rupiah. However, this policy is not good in the long run because an increase in the benchmark interest rate can repeatedly cause a decrease in GDP. This is because people will tend to keep their money in the bank because it gives a good interest return instead of using their money for business activities due to high credit interest. In the end, this policy was ineffective, because to increase foreign investment activities in Indonesia would weaken economic productivity in the country.

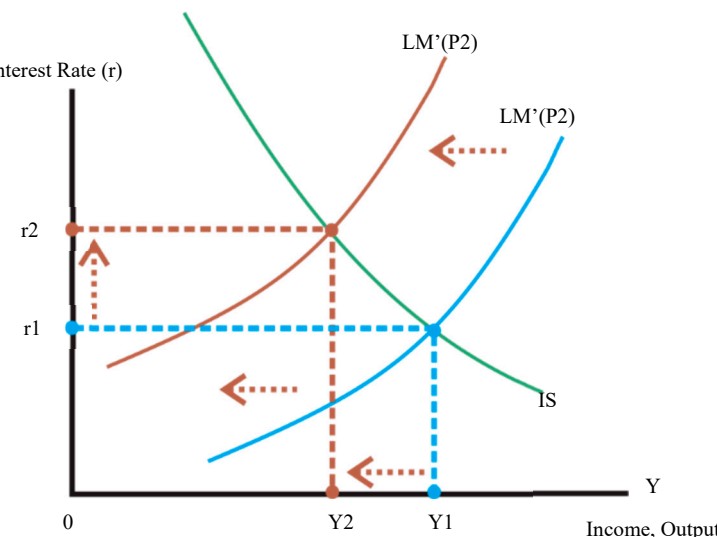

**Figure 5.** Money market and goods market movements (IS-LM curve).

### 4.2. Effectiveness of Fiscal Policy

Referring to the GDP equation as follows Y = C + I + G + NX (E-I), the institutional role of the government in responding to future economic growth that is projected to decline due to the global economic downturn is to increase investment. This is because other influential variables on GDP such as consumption, government spending, and net exports are not effective in increasing GDP when faced with capital outflow conditions due to declining global economic conditions and trade wars. When domestic production activities are sluggish, the goods produced also decline. The public will not be encouraged to consume because of expensive imported goods and sluggish production activities in the country.

Viewed from the variable government expenditure, increasing government spending will indeed increase GDP but further increase the current account deficit because the government will use more budget for shopping, and import activities will be even greater. To regulate exports and imports in the midst of the American and Chinese trade wars and the global economic downturn, it will be difficult for the government to reduce the current account deficit because reducing the deficit means lowering GDP growth. Indonesia needs to increase export activities outside of China and America to increase savings while simultaneously reducing the level of public consumption. On the other hand, imports as a step to support domestic investment will be difficult to suppress because Indonesia is a country that still relies on imported goods. In the end, reducing the current account deficit will not be effective because Indonesia requires a combination of high savings and low investment while the actual conditions that occur are not appropriate.

Therefore, the right fiscal policy is to increase the flow of investment from outside to Indonesia through foreign direct investment, which is used to increase investment activities in the country that provide GDP growth (Godfrey et al. 2016; Rowlinson and Hassard 2013; Scott 2014). Fiscal policy through foreign direct investment is currently not effective because it turns out that Indonesia is financing the current account deficit with volatile capital flows from investor portfolios so that often economic turmoil makes investors quickly withdraw their investment out of Indonesia and capital outflows occur. This condition makes Indonesia volatile because often the incoming investment flow is hot money. Whereas the real goal is to obtain capital inflows such as export-oriented FDI because it is more stable and is not easy to get in and out of Indonesia due to economic turmoil. In addition, FDI will create jobs in Indonesia so as to encourage economic growth.

### 4.3. Post-Trade War Investment Opportunities in Indonesia, Vietnam, and Cambodia

The trade war between America and China resulted in American companies operating in China to take alternative steps by moving their manufacturing base out of China (Yu et al.

2016). The plan is a big opportunity that needs to be captured by countries in the Southeast Asian region. The following are some examples of case studies of several countries in the ASEAN that took advantage of the trade war. In order to capture the post-trade Indonesia investment opportunities, Indonesia, in carrying out its institutional role, needs to reform its policies by prioritizing aspects of credibility, certainty, and compliance aspects, as well as carrying out the oversight function of implementing institutional policies to create policies that synergize and provide certainty and convenience for foreign investors.

Indonesia has a need to increase the share of export-based FDI that can reduce the current account deficit because investment flows from American companies to Indonesia will increase GDP, provide new jobs for the Indonesian people, develop Indonesia's manufacturing industry, and increase Indonesia's technology.

Furthermore, export volumes in future Indonesia can go up because the products produced by foreign companies have definite selling power in the international market. Indonesia, as a country rich in natural resources, should be a point of excellence that can be glimpsed by investors as the choice of country to move their manufacturing base to from China.

However, the facts show the opposite. Indonesia is actually unable to compete with other countries in Southeast Asia such as Cambodia, Vietnam, Malaysia, and Thailand. The following (Figure 6) is the flow of FDI to countries in Southeast Asia.

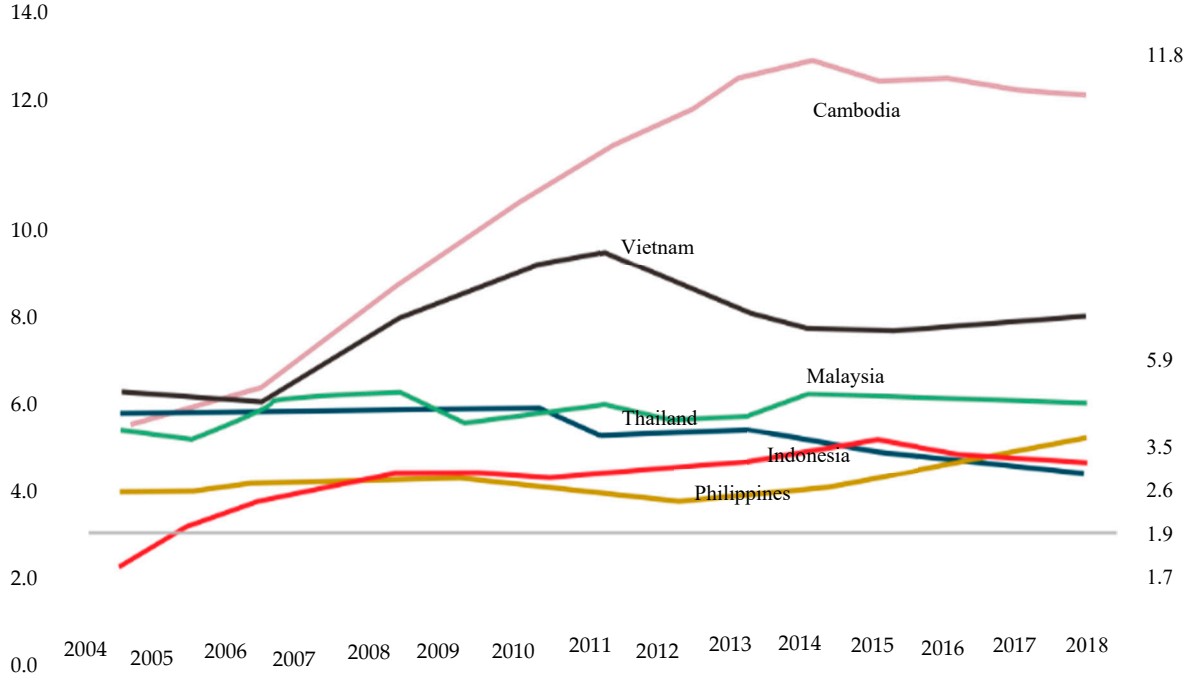

**Figure 6.** World Bank and World Development Indicators FDI in the ASEAN 2004–2018. Foreign direct investment (FDI) net inflows percentage of GDP, 5-year moving average.

Businesses moved out of China but did not go to Indonesia. Based on FDI flows recorded by the World Bank, two countries with the highest FDI flow were occupied by Cambodia and Vietnam. The following will explain the advantages of the country to capture the interest of investors to invest in the country.

### 4.4. Investment Opportunities of Vietnam Post Trade War

Vietnam is a country bordering mainland China, so territorially it is a strategic country to be the destination of moving a manufacturing base from China. Although Vietnam is not as advanced as China, several factors below are key strategies that have seen Vietnam being selected as one of the preferential countries by investors. Vietnam has a corporate tax rate of 20%, while Indonesia is still at a rate of 25%, except for companies that have gone

public, which are subject to a corporate tax rate of 20%. In addition, Vietnam has imposed another 3% tax tariff cut for disadvantaged regions and 10% for heavily disadvantaged regions. This is not yet owned by Indonesia, so an evaluation of fiscal policy needs to be conducted by Indonesia to attract foreign investment into the country.

The ease of starting a business in Vietnam, according to World Bank data, is ranked 104th, while Indonesia ranks 134th. The ease of licensing to start a business is supported by permit procedures that only cover two types of permits, namely, investment approval (including import licenses and the use of foreign workers), which is published no later than three days, and business registration (including export and import interests and sources of procurement of raw materials), which is published no later than one month. The two licenses that are more concise, given the impact of the commercial production process in Vietnam, only take approximately 6 months. In addition, Vietnam has centralized one-door licensing so that investors only deal with one authority. Other indicators assessed are the amount of costs incurred and the minimum paid up capital.

Vietnam has a logistical performance index ranking (39) which is better than Indonesia (46) based on World Bank data for 2018. This is important because logistics is a factor that will support export and import activities in a country. Vietnam has a broader level of diversification of the manufacturing industry ( Machinery, Garments, Footwear, and Others) compared to other neighboring countries so that investors can seek efficiency by making choices for the number of businesses and factories producing similar products. Table 2 shows the results of Vietnam's leading exports in USD for 2019.

**Table 2.** Top exports of goods in Vietnam (USD millions), 2019.

| |
|---|
| Machinery |
| 133,135 |
| Garments and textiles |
| 36,926 |
| Footwear, headwear |
| 24,845 |
| Food and beverage |
| 22,793 |
| Metals |
| 10,381 |
| Others |
| 62,315 |

Source: Author's calculation using Vietnam trades and index (World Bank 2019c).

The existence of export specialization in the field of machinery has increasingly attracted manufacturing investors to reallocate from China to Vietnam because machinery is very closely related in the building and manufacturing industry. Based on trading across borders, Indonesia (rank 116) is still below Vietnam (rank 100), which is judged from supporting aspects of cross-border activities such as aspects of border compliance, documentation, and domestic transportation. Vietnam has an easy-to-get credit index (rank 32) compared to Indonesia (rank 44) in terms of lending and borrowing procedures and credit registration.

This will have a positive impact on investors entering Vietnam because credit facilities will provide a source of funds that makes it easier to expand manufacturing. Vietnam has relatively low labor costs so that investors can be more efficient. In addition, Vietnam has a human capital score 62.19 that is not too much different from China 67.72. These two factors make the principle of business substitution stronger (Table 3).

**Table 3.** Human capital in Vietnam and China.

| Country | Monthly Minimum Wage | Human Capital Score (0 to 100) |
|---|---|---|
| Vietnam | USD 125–180 | 62.19 |
| China | USD 140–346 | 67.72 |

Source: Author's calculation (World Bank 2019c).

Vietnam's foreign direct investment (FDI) has continued to grow since 2014 based on the following data in Figure 7.

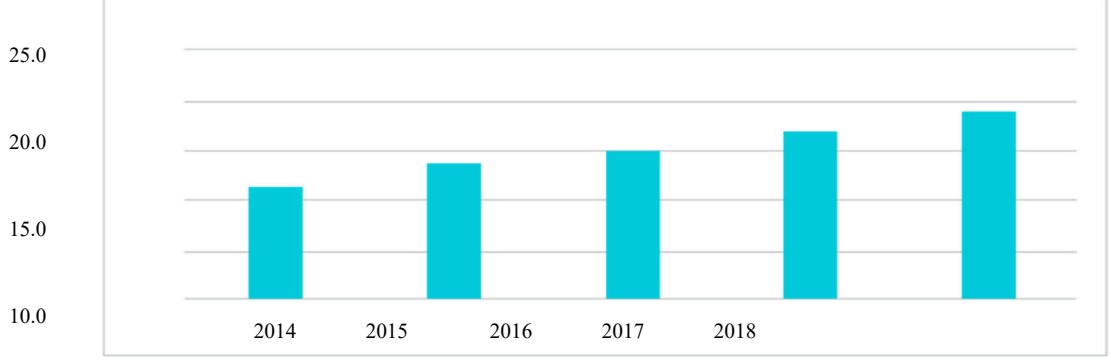

**Figure 7.** Foreign direct investment in Vietnam in the years 2014–2018.

*4.5. Post-Trade Cambodia Investment Opportunities*

As a country directly bordering with Vietnam, Cambodia is also one of the alternative countries chosen by investors to move its manufacturing base from China. Over the past 10 years, Cambodia's FDI has risen more than 800% with its highest achievement in 2018 of USD 3 billion. This is supported by several policies, namely: (i) Flexible currency policy where 80% of transactions in the country use USD. This certainly makes it easy for foreign investors to do business because there is no need to think about exchange rates against currencies. (ii) The Cambodian property market is open for foreigners to make investments without changing citizenship. In Cambodia, 51% of the property is owned by Cambodians and 49% is owned by foreign parties. In addition, foreigners have land use rights of up to 50 years, so that the opportunity to establish a manufacturing base is easy and profitable. Cambodia also extends its 100% ownership policy to companies that are based there, so the opportunity to open up business is very exciting for investors. (iii) Foreign parties that register their investment projects with the Council of Development of Cambodia (CDC) will be given a tax cut of up to 40% on material production (Shihlun 2018). The tax exemption (tax holiday) in force in Cambodia for three years provides an attractive stimulus for investors to invest in the country. After 3 years of tax exemption, foreign investors can have priority over an extended period of three years later depending on the type of project, and the amount of capital invested. (iv) Cambodia has a special economic zone that gives privileges to industries that contribute FDI to the country. The rights granted include the assignment of state officials who are ready to help investors, and the provision of assistance for infrastructure and public works to support industry.

*4.6. Institutional Change and Trade War Phenomenon in the ASEAN*

The ASEAN as a community institution of several countries has the authority to formulate appropriate policies in responding to the ongoing economic conditions. In addressing these matters, ASEAN countries, which in this case are the government, must place institutional policies as the basis for taking appropriate actions in order to achieve economic sustainability. In responding to the influence of trade war and reflecting the loss of business opportunities that enter ASEAN countries, as a state institution, reforms to institutional policies need to be considered because they have an impact on institutional

changes. The following are the factors impacting on how future economic expectations ultimately influence government decisions on institutional policy reform (Rodríguez-Pose 2013). (i) The period of failure of the current policy has led ASEAN countries to the decline of the old paradigm and gave birth to opportunities for the search for new solutions. Although the effects of trade war did not directly impact the economic crisis in ASEAN countries, there was economic uncertainty during this period. (ii) New paradigms emerge and offer policy solutions that are recommended from the epistemic community and implemented by politicians in Indonesia. The basis of the success of this step requires commitment from the government not only as a policymaker but as an institution that monitors future policy steps in order to assess the effectiveness of these policy reforms. Independence and transparency are important factors needed to maintain policy reliability. As a step to capture opportunities for trade war, the current policy paradigm is that it no longer relies solely on raising the benchmark interest rate as a stimulus to monetary policy and reducing taxes as a stimulus to fiscal policy, but rather the opportunity for capital outflows flowing from China to countries in the ASEAN shows a solution and new effect on reducing the current account deficit in ASEAN countries. As a result, the government has an important role to change the paradigm as a policy solution that updates the current policy. (iii) Institutional opportunities in imitating the policies of other countries and instilling a new paradigm as an institutional framework in the ASEAN. Countries in ASEAN who have lost interest in attracting foreign investors to invest domestically make lessons for the level of institutions to benchmark the excellence of other ASEAN countries.

This certainly creates opportunities for free flow of investment in the search for the destination country. The ASEAN still finds hot money investment flows cannot depend entirely on efforts to reduce the current account deficit. This is because volatile investment flows will be very easy to move out when the macroeconomic conditions of the ASEAN countries are not good. Therefore, reliable instruments such as FDI are needed that provide certainty in the long run in order to increase GDP. The following is the exposure of financial assets to GDP of other ASEAN countries based on data from the World Bank (Figure 8).

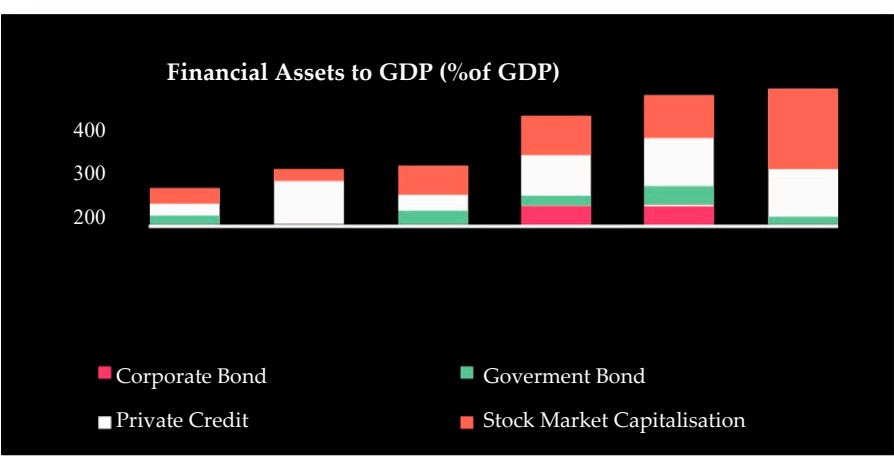

**Figure 8.** Product financial assets to GDP in the ASEAN, 2019.

The need for FDI and FDI opportunities that arise as a result of trade war should be able to be utilized more optimally by ASEAN countries. However, this was not captured because the flow of investment moved towards only other ASEAN countries such as Vietnam, Cambodia, and Malaysia. The most significant factor of failure is due to inefficient and effective procedures and policies so that policy reforms need to be formulated.

The government as an authorized institution has an important role in responding to this phenomenon so that it is expected to make institutional policy reforms to provide the stimulus to macro variables that lead to an increase in GDP and decrease the current account deficit. The following is the role of a government as an institution in driving policy reform: (i) Credibility aspects in order to emphasize Indonesia's position as a country that

is open to business opportunities. This can be achieved by: (a) integration into the supply chain. The structure of the global supply chain will help Indonesia and the Philippines in getting support and recognition from other countries, thereby increasing mutual relations between countries. Imports and exports need to be accommodated by the flexibility of the policy by prioritizing the goal of increasing FDI; (b) paving the way for foreign investors to enter Indonesia and the Philippines through relaxing the limits of foreign equity on the list of negative investments that play a key role; and (c) enabling investors to recruit the right aspects of expertise in order to develop the business. This can be achieved by reducing the requirements for professional-based worker qualifications. (ii) Certainty aspects that reduce discretion and regulative uncertainty to be able to convince investors. Steps that can be taken are removing contradictory, inconsistent policies, discretionary policies related to licensing and business registration, creating instruments to oversee conflicting regulations between regional and central regulations, evaluating legal and regulative policies related to exports, and investments that consider costs and benefits for business, society, and government. The following are facts from data collected by the World Bank regarding aspects of institutional policy certanty to investors: (a) during 2015–2018, there were more than 6300 government policies that were issued (representing 86% of government policies) when in 2011–2014, they represented 82% or as many as 5000 policies; (b) there is no single entity that can be relied on to ensure that the policy has served the priorities of the government because each institution has a limited role in terms of policy coordination and review; and (c) in 2017, local regulations contradicted and 61% of those regulations related to business fees and permits. (iii) Compliance aspects of the government's strategy as an institution. This can be accomplished by empowering the formation of a team responsible for the government to identify arbitrary administrative policies and regulations that contradict government regulations and laws that will affect investors and impose consequences on authorities who deviate from the direction of government policy.

### 4.7. Game Theory Application and Government as a State Institution

Departing from the presentation of policy reforms that must be carried out by the government, the dilemma occurs when the government which has an institutional role is faced with its credibility related to the policy reforms. The steps that can be formulated are by looking at the decisions of the central government and regional governments in the perspective of game theory.

The role of government as a state institution must be mutually sustainable between the central government and regional governments. The central government has a centralized function that holds the fostering and supervision functions of the administration of regional government affairs, while the regional government has a more specific function in terms of regulating the people in their regions. To maintain sustainability, there is a need for communication, coordination, and interaction between the central government and regional governments that support the implementation of institutional policies in the ASEAN.

When permitting business becomes inefficient, the response must first be captured by the local government as the institution that carries out policies in its autonomous region. This is because the regional government acts as an intermediary between the central government and the community in channeling the aspirations of the community and disseminating policy from the central government. In addition, the regional government also acts as an institution that will carry out public policies and report the effectiveness of these policies on the sustainability of regional autonomy to the central government.

Continuing the function of local government, the current reform of public policy can be initiated by the regional government itself (the local government plays an active role) and public policy is only followed by the regional government based on the policy of the central government without any input (the local government plays a passive role).

The existence of notes from the World Bank regarding the uncertainty of policies and contradictory policies indicates that the central government and regional governments have not agreed on the policies that govern business licensing procedures in Indonesia

([World Bank 2019a](#)). Local governments that do not play an active role in voicing conditions on the ground because they are vulnerable to change produce "follow" action steps with a payoff value of 0. Meanwhile, the initiation steps obtain a payoff value 1 because doing the initiation requires an analysis of the scope of the problem, enthusiasm for change and development, and good communication between the regional government and the central government.

This also applies on the side of the central government when the central government does not respond to input from the local government and chooses to stick with the current policy, and the action "follows" gives a payoff value of 0. However, when the central government initiates a good step based on input from the local government and based on the results of supervision by forming an independent body, the initiation step will provide a payoff value of 1 (Figure 9). Here is a game table that combines decisions from the central government and local governments based on the value of the payoff that arises.

**Figure 9.** Author simulation of game theory application and government as a state institution—initiate and follow 0 to 1.

Based on the game table above, the best response from the central government is to choose to initiate without being affected by any decisions from the local government because the initiation action will give a payoff value of 1 (blue). The best response from the regional government was also initiated to provide input to the central government regarding policies/regulations that are currently running (yellow). By seeing the best response from the decision of the central government and regional governments, a balanced equation is obtained where the best response to the sustainability of institutional policies in Indonesia is to mutually initiate policies from both the central and regional governments so that the policies formulated can be in accordance with Indonesia's development goals and aspects of certainty and compliance can be achieved. The central government can carry out regular oversight and coordination functions with regional governments to solicit input regarding the effectiveness of policies implemented in each regional autonomy so that these inputs can be used to assess regulations and make improvements to be more targeted and effective.

Meanwhile, local governments can routinely carry out the reporting function in order to provide input related to real conditions on the ground so that they do not contradict state regulations. Nash's equilibrium is reached in column 1, row 1 (bold). After coordination in formulating/revising of institutional policies is carried out, the next most important part is how to maintain continuity between the commitment of the central government and regional governments in implementing the policy. The existence of contradictory policies is evidence of the discontinuity that is illustrated through the following game theory (Figure 10).

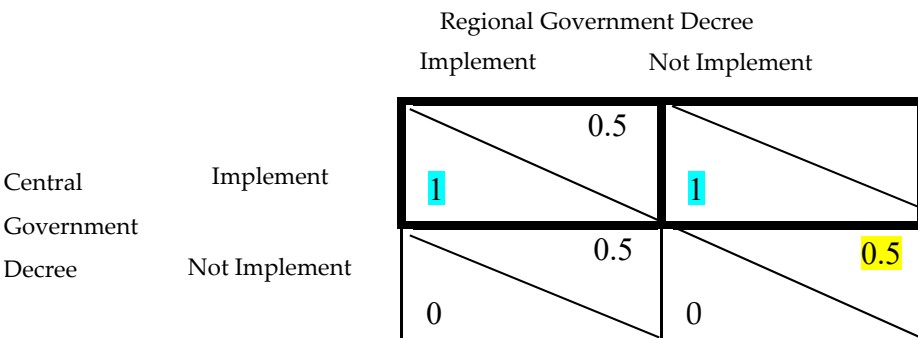

**Figure 10.** Author simulation of game theory application and government as a state institution—implement and not implement 0.5 to 1.

When the local government decides not to implement policies according to the direction of the central government, this shows that the credibility of the local government becomes questioned, and that the value of the payoff given for each decision of the local government becomes 0.5 because local governments can change their decisions based on their own considerations.

The results formed from the game theory above produced two variations of responses, namely, the decision of the central government and regional governments to jointly implement institutional policies, and the decision of the central government to carry out institutional policies, but the regional governments did not carry out institutional policies. Game theory becomes very dependent on the response of the local government, which ultimately results in a gray equilibrium because the credibility of the regional government is questioned, as is the power of the local government in determining the sustainability of institutional policies.

Therefore, intervention from the central government is needed to form an independent supervisory body that functions to oversee arbitrary and contradictory policies so that follow-up in the form of consequences can be given to local governments that deviate from the direction of institutional policies. The existence of these interventions will change the game theory to be as follows (Figure 11).

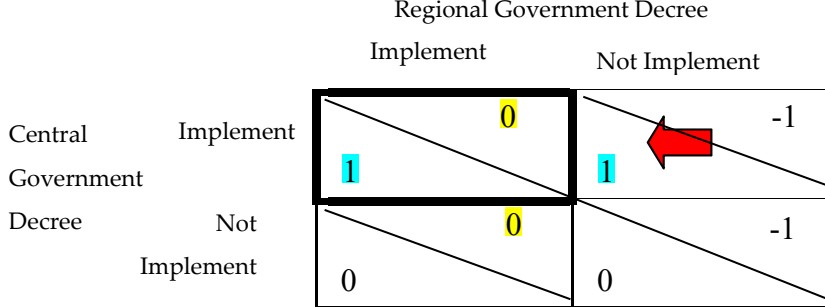

**Figure 11.** Author simulation of game theory application and government as a state institution—initiate and follow −1 to 0 and 1.

Nash's equilibrium generated through intervention from the central government is resulting in compliance with local governments because local governments will not dare not to implement institutional policies regarding the consequences imposed by the central government so that the payoff value for not running is −1. Eventually, the equilibrium point is formed in column 1, row 1, with the total payoff value of 1.

## 5. Conclusions and Recommendations

Based on the discussion on the effect of trade war and the role of institutional policies in Indonesia, the following conclusions can be drawn:

Monetary policy, by raising the benchmark interest rate to strengthen the attractiveness of domestic financial assets in ASEAN countries in the eyes of investors, is effective in the short term to maintain the attractiveness for investors to invest in the ASEAN and minimize capital outflows.

Fiscal policy, in order to increase GDP and reduce the current account deficit, is to increase foreign direct investment into the country so that it can increase GDP, provide new jobs for the ASEAN community, develop industry manufacturing in the ASEAN, and increase the technology base in the ASEAN so that the export volume in ASEAN countries increases.

Factors causing foreign companies to leave China and not invest in several ASEAN countries (Indonesia and the Philippines) are: (i) procedures governing foreign investors to open business opportunities in terms of cost and time efficiency; (ii) imports are strictly regulated, such as the imposition of import tariffs for production inputs, the National Certification Agency by third parties, import licenses, and the existence of multilevel inspections for several types of production inputs; (iii) human resources that are felt to still need improvement in the quality of their work; and (iv) policy discretion between the central government and regional governments, which creates regulatory uncertainty about foreign investment.

The government needs to fix institutional policies that pay attention to aspects of credibility to show integrity and commitment in accommodating the needs of foreign investors, certainty aspects by eliminating institutional policies that are contradictory, mutually discrete, and uncertain, and compliance aspects by carrying out an independent oversight function which is followed up with an independent persuasive approach and a consequence approach.

Recommendations that can be made based on the above topics are institutional policy reforms through benchmarking to ASEAN countries such as Indonesia, Vietnam, and Cambodia need to be carried out. Evaluate and improve the ability to exist in human resources in specific sectors through the empowerment of human resources by the government. In the case of constitutional policy, independence and transparency to maintain policy reliability is needed. In terms of monetary policy, it is necessary to determine the period in terms of increasing the benchmark interest rate because it can cause a decrease in GDP if undertaken repeatedly. In the case of fiscal policy, a comprehensive budgetary plan can be carried out, namely, by reducing taxes that aim to stimulate economic growth and attract investment.

The purpose of this article was to explore in-depth how the institutions that ASEAN countries Indonesia, Vietnam, and Cambodia have built in the face of the trade war between China and the USA to obtain input and policy recommendations. This article sees that the ASEAN countries of Indonesia, Vietnam, and Cambodia still have coordination line problems in institutions, human resource issues, budget issues, institutional arrangements, and ideal division of tasks that can optimize the direction of the country's policies. This article provides significant benefits for the countries that are the sample of this research as a reflection to improve institutions, both for the ASEAN community and other issues.

**Funding:** This research received no external funding.

**Institutional Review Board Statement:** Not applicable.

**Informed Consent Statement:** Not applicable.

**Data Availability Statement:** Not applicable.

**Conflicts of Interest:** The author declares no conflict of interest.

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
