# Peer review of "Institutional Change and Macroeconomic Variables in the ASEAN—Indonesia, Vietnam, and Cambodia: The Effects of a Trade War between China and USA"

_economies, doi:10.3390/economies9040195_

Round 1

Reviewer 1 Report

in my opinion, it is necessary to clarify the title of the article. The article does not cover all ASEAN countries, so I would specify " the example of Indonesia, Vietnam and Cambodia"

Author Response

Dear Reviewer 1,

Here I submit a response to the improvement of my article the you commented on. Attached document, feedback for reviewer 1 comments. 

Best,

Fransiskus X Lara Aba 

Reviewer 2 Report

The article entitled „Institutional Change and Macro-Economic Variables and the Effects of Trade War in ASEAN Countries” seems to be interesting.

Unfortunately there are some mistakes, which ought to be corrected:

1) The article’s title this not contain any information between what countries the trade war is going on. I suggest changing the title into: „Institutional Change and Macro-Economic Variables in chosen ASEAN countries during the years….. ”. Some additional information about this trade war ought to be given in the article.

2) The article structure is a little bit confused. The theoretical parts intertwine with empirical (for example fig. 1, 5) . The Game theory ought to be located in theory part. You can divide the theoretical part into three sections:

Literature review about:

-Institutional Changes,

- Macro-Economic Variables and

- the Game Theory in economy and management.

3) Introduction, page 1: It is written (Alicia, 2020) but it ought to be (Herrero, 2020). Please change it in the references as well. Alicia is the name, not the surname.

4) On page 2 the author has pointed out that “The stages of public policy making according to W. Dunn are as follows…”. Please indicate why You have chosen W. Dunn and his proposal of stages.

5) Page 2: there is no permission to find fiscal policy as the same as Keynesian policy [(i) Fiscal (Keynesian) Policy] and Monetary policy as Monetarist [(ii) Monetary Policy (Monetarist)]. It is so, because the economy practice has been changing (under the influence of the globalization processes and the IT revolution). All these changes have influenced the way the macro-economical dependences are understood and have also contributed to the shift in the theoretical re-evaluation for example in macro-economical policy as well as other fields.  The views on the monetary policy as well as the fiscal policy have evolved  and many times being far from the foundations of the monetary and Keynesian schools.  

6) Page 3: The author has written “The fiscal policy is based on the theories of the British economist J.M. Keynes…”. Is the fiscal policy based only on the foundations of Keynes? What about the views such schools as: new classical economics, Real business-cycle theory (RBC theory), The Austrian School, New institutional economics (NIE), New political economy, etc.?   

7) Page 3: The author has written “(ii) Contractional fiscal policy”. Please check this words in terms of spelling.

8) Page 3: My remark refers to the four components of the fiscal policy. The first two are clear but the rest requires more explanations. Fiscal policy refers to above all taxes and state expenses and to managing public assets to gain control and to influence the division of income as well as the general level of economy activity of a given country. FDI activity is treated rather as the part of widely understood economic policy and not as strict fiscal policy. A similar remark concerns the component of Debt/Surplus Management. It should be rethought if this is a component of this policy or rather it is the result of it.

9) Page 4: In the theoretical part the author presents the empirical examples concerning Indonesia. In my opinion the theoretical part of the article should be clearly divided from the empirical one. 

10) Page 4: In the references (Thornton, 2001) please reconsider the style of this sentence. Remember that Patricia is the name similarly to Malcolm and not a surname.

11) Taking into consideration the methodical approach and assumptions concerning the analysis of three ASEAN countries (Indonesia, Vietnam, Cambodia) I am of the opinion that there should be references and data referring to all of them. It concerns: table 1, fig 2, 3,4, table 2.

12) The remarks concern the titles of tables and figures:

- table 1: there is no information which year(s) this data concerns.

- figure 2: Projected decline in GDP 2010-2022? The data concerning the year 2010 cannot be projected because they are a fact. Now we live in 2021. You must give information in the title of this figure what country the data refers to.

- Figure 3: You cannot title this figure: “efficiency of monetary policy”. There is only a presentation on current account balance not the whole efficiency or the lack of it. Please add the information about what countries and year(s)) it refers to.

- Figure 4 is titled “Indonesian imports and exports”. Please add the years and the name of currency.

- Numeration of figure 5 is doubled,

- There is no proper name of fig 5, page 10: Indicators? Specify the name of indicator and give the information about the units.

- The title of table 2 is not proper. It ought to be: Top exports of goods in Vietnam in…. year.

- Fig 6: Foreign direct investment in Vietnam in the years ….. [ units].

- The title of Fig 7: there is no information on the years this figure concerns. The values of the horizontal lines are not given.

- Page 16 (one figure), 17 (two figures). There are no titles of these figures.

13) The separated part of this article entitled” Conclusions and recommendations” ought to be prepared. So far the conclusion has been done, but please find some recommendations for Indonesia, Vietnam and Cambodia institutional policy and for ASEAN as an

14) Please indicate the main goal of this article, the research questions or hypotheses and try to refer to them at the end of the article.

I hope that this remarks, comments will allow the author to prepare a better article.

Author Response

Dear Reviewers 2

Here I submit  a response to the improvement of my article that you commented on.  

Point 1) The article’s title this not contain any information between what countries the trade war is going on. I suggest changing the title into: „Institutional Change and Macro-Economic Variables in chosen ASEAN countries during the years….. ”. Some additional information about this trade war ought to be given in the article.

Response for Point 1): I agree with the input, as also suggested by reviewer one so that the title is changed, I will change the topic of the article with a new title, namely; "Institutional Change and   Macro-Economic Variables in ASEAN – Indonesia, Vietnam And Cambodia the Effects of Trade War between China and USA."

While additional information about the trade war that I added in the introductory chapter, namely: The trade war is a term used by media and repeated by politicians and economists for United States (US) action who have raised import duties imports of manufactured products China and some other countries and got a backlash from destination country. Action against China is taken because the US considers the country has jeopardized the interests US national. China's export volume to the United States far exceeds the proportion of imports so that US trade deficit with China since 2010, increasingly increase. To reduce the deficit trade, since June 2018 US government launches strategy increase import duties, especially towards China. On early May 2019 US Government increase tariffs on imports Chinese commodity worth 200 billion US dollars from 10% to 25%.

China responds by imposing new import tariffs as of June 1, 2019, for items belonging to US 60 billion US dollars. China too is considering stop purchasing the product agriculture and Boeing aircraft from The US. This trade war is entering a new chapter when the company Google from the US decided android system connection with Chinese mobile phone manufacturers, Huawei in response to government prohibition. Ban this has an impact significant for product users Huawei. Hundreds of millions of users Huawei phones can't anymore access exclusive services Google-like Gmail, Youtube, play store, and google maps. US mobile parts company, Lumentum has also decided to stop sending spare parts to Huawei. Intelligence The US believes Huawei is supported by the military so that it is suspected provide intelligence services for Chinese government.

Point 2) The article structure is a little bit confused. The theoretical parts intertwine with empirical (for example fig. 1, 5) . The Game theory ought to be located in theory part. You can divide the theoretical part into three sections:

Literature review about:

-Institutional Changes,

- Macro-Economic Variables and

- the Game Theory in economy and management.

Response for Point 2): I agree to change it, as for the changes made are as follows:

2.1. Public Institutional Change

2.2. Macroeconomic Stabilization and Variable Policies.

  1. 3. The Game Theory in Economy and Management

This game theory aims to analyze the decision-making process of different competitions and involve two or more Countries or company interests in running their business. The usefulness of a game theory is the methodology it provides for structuring and analyzing strategy selection problems. Using game theory, the first step is to explicitly determine the players' existing strategies and determine each player's preferences and reactions. Two game strategies can be used in-game theory: pure approach (each player uses a single system) and mixed strategy (each player uses a mixture of different methods). The refined design is used for games whose optimal results have a saddle point (a kind of balance point between the game values ​​of the two players). Meanwhile, the mixed strategy is used to find the optimal solution for the game theory case that does not have a saddle point.

Some several essential elements or concepts are crucial to solving each case with game theory. Here's a full explanation:

a). Number of Players

Games are classified according to the number of interests or goals that exist in the game. In this case, it should be understood that the notion of "number of players" does not always mean the same as the "number of people" involved in the game. The number of players here means the number of groups or countries based on their respective interests or goals. Thus, countries or companies that have the same interests can be counted as a group of players.

b). Reward / Pay-off

Rewards/pay-offs are the final results that occur at the end of the game. Regarding these rewards, games are classified into two categories, namely zero-sum games and non-zero-sum games. ). A zero-sum game occurs when the sum of the tips of all players is zero, i.e., by taking each gain as a positive number and each loss as a negative number. Apart from that, it is a game of numbers – non-zero. In zero-sum games, every win for one player is a loss for the other player. The significance of the difference between these two categories of reward-based games is that zero-sum games are a closed system. At the same time, the non-zero-sum game is not like that. Most games are essentially zero-sum games. Various situations can be analyzed as a zero-sum game.

c). Strategy Game

Game strategy in game theory is a strategy or a specific plan from a country or company as a player, as a reaction to actions that may be taken by players who are rivals. The games are classified according to the number of strategies available to each player. If the first player has m possible strategies and the second player has n possible strategies, the game is called an m x n game. The importance of the different types of games based on the number of strategies is that the game is divided into finite games and infinite games. A little game occurs when each player's most significant number of strategies is limited or specific. In contrast, an endless game occurs when at least one player has an unlimited or indefinite number of strategies.

d). Game Matrix

Every game analyzed with game theory can always be presented in the form of a game matrix. The game matrix is ​​also called the reward matrix, which is a matrix in which all the elements are rewards from the players involved in the game. The rows represent the strategies of the first player, while the columns represent the other players' strategies. Thus, the m x n strategy game is denoted by the m x n game matrix. Game theory assumes that the system available to each player can be calculated, and the rewards associated with them can be expressed in units, although not necessarily in monetary units. This is important for the game's completion, namely, determining the choice of strategy that each player will carry out, assuming that each player tries to maximize their minimum profit. The value of a game is the average reward / expected reward throughout the game, assuming both players always try to play their optimal strategy. Conventionally, the value of the game is seen from the side of the player whose strategies are represented by the rows of the reward matrix; in other words, seen from the point of view of a particular player. A player is said to be fair if the value is zero, where no player gains an advantage or victory in an unfair game (unfair), a player will win over another player, i.e., if the value of the game is not zero, in this case, the player's value is positive if the first player (row player) wins, otherwise, the game value is negative if another player (column player) wins.

This is where the role of game theory is to determine which strategy will be decided by policymakers in a country to seize the market. The competition exemplified above can be identified to explain the concept of game theory which consists of several essential elements, namely:

  1. The numbers in the pay-off matrix, commonly called the game matrix, show the results (pay-offs) of different game strategies; these results are expressed in the form of practical measures such as money, percentage market share, or utility.
  2. The maximizing player is the player who is in the row and who wins/gains the game, while the minimizing player is the player who is in the column and suffers a loss/loss.
  3. Game strategy is a series of activities or a comprehensive plan of a player as a reaction to the behavior of his competitors. In this case, the strategy or program cannot be undermined by other competitors.
  4. The rules of the game are the patterns in which the players choose a strategy.
  5. The value of the game is the pay-off result estimated by the player throughout the series of games in which each player uses his best strategy. The game is said to be fair if the value of the game is equal to zero and vice versa.
  6. Dominance is a condition where the player with each pay-off in the strategy is superior to each related pay-off in an alternative method. The dominant rule is used to reduce the pay-off matrix size and computation effort.
  7. Optimal strategy is a condition wherein a series of game activities, a player is in the most advantageous position regardless of the needs of his competitors.
  8. The purpose of the model is to identify the optimal strategy or plan for each player.

Point 3) Introduction, page 1: It is written (Alicia, 2020) but it ought to be (Herrero, 2020). Please change it in the references as well. Alicia is the name, not the name.

Response for Point 3): Changes to (Herrero, 2020), including in the reference Herrero Alicia Garcia (2020) Europe in the midst of China-US strategic economic competition: what are the European Union's options? Journal of Chinese Economic and Business Studies.

Point 4) On page 2 the author has pointed out that “The stages of public policy making according to W. Dunn are as follows…”. Please indicate why You have chosen W. Dunn and his proposal of stages.

Response for Point 4): Public Policy Analysis (William N. Dunn, 1994): An Applied Social Science discipline that using a variety of research methods and arguments to generate and transform relevant information with the policies used in specific political environment to solve policy issues related to the trade war due to the process This trade war takes place to head, so it's hard to expect can be resolved bilaterally shortly, however The world community has felt its impact. These impacts include, among others, the exchange rate of several countries that began to depreciate and the value of stocks. The economic conditions of ASEAN countries: Indonesia, Vietnam, and Cambodia, as well as the world, will be affected by the trade war, which will ultimately affect the domestic economy of countries in the world that have trade relations with China and the USA.

Point 5) Page 2: there is no permission to find fiscal policy as the same as Keynesian policy [(i) Fiscal (Keynesian) Policy] and Monetary policy as Monetarist [(ii) Monetary Policy (Monetarist)]. It is so, because the economy practice has been changing (under the influence of the globalization processes and the IT revolution). All these changes have influenced the way the macro-economical dependences are understood and have also contributed to the shift in the theoretical re-evaluation for example in macro-economical policy as well as other fields. The views on the monetary policy as well as the fiscal policy have evolved and many times being far from the foundations of the monetary and Keynesian schools.

Response for Point 5): Agree to abolish Keynesians and Monetarists because both fiscal and monetary economy is economic theories presented by Keynesians regarding total expenditure in fiscal and monetary policy. Keynes advocated increasing government spending and lowering taxes to stimulate demand and pull the global economy out of depression.

Point 6) Page 3: The author has written “The fiscal policy is based on the theories of the British economist J.M. Keynes…”. Is the fiscal policy based only on the foundations of Keynes? What about the views such schools as: new classical economics, Real business-cycle theory (RBC theory), The Austrian SchoolNew institutional economics (NIE), New political economy, etc.? 

Response for Point 6): In the literature review study, I will add questions from Reviewer 2 regarding the concept of fiscal policy from new classical economics, Real business-cycle theory (RBC theory), The Austrian School, New institutional economics (NIE), New political economy.

Policymakers should ensure automatic corrections occur rather than engage in active fiscal and monetary policy on the concept of policy from theory. The new classical economists' policy suggestions are labor market reform, innovation through entrepreneurship, and ensuring the economy operates under a free market. These are all important for influencing aggregate supply. As a result, RBC theory rejects Keynesian economics or the view that in the short-run economic output is primarily influenced by aggregate demand and monetarism, a school of thought that emphasizes the role of government in controlling the money supply. Despite their rejection of the RBC theory, today's two schools of economic thought represent the foundation of mainstream macroeconomic policy.

The Austrian School believes that misguided government intervention causes business cycles. To boost economic growth, the government cut interest rates to artificially lower levels. Interest rates that are too low cause companies to accumulate too much equipment. That shifts aggregate demand to the right and creates an inflation gap in the economy.

NIE explores the notion of non-market institutions (property rights, contracts, revolutionary parties, etc.) to compensate for market failures. In the NIE approach, imperfect information, production externalities, and public goods are identified as the most important sources of market failure, thus necessitating the presence of non-market institutions.

On the other hand, in the neoclassical approach, the three variables above are assumed not to exist, so the transaction costs associated with these variables are considered non-existent. In addition, the NIE literature also adds several essential points about institutional failures that are the cause of underdevelopment in many countries.

The New political economy approach is defined as the relationship between political aspects, processes, and institutions with economic activities, such as production, investment, price creation, trade, consumption, etc. Based on this definition, it can be concluded that political economy relates all political administration to economic activities carried out by the community and those introduced by the government. Economics and political science are different, but they can be compared, given that they have the same process.

Point 7). Page 3: The author has written "(ii) Contractional fiscal policy." Please check these words in terms of spelling.

Response for Point 7): That's right, I changed the word Contractional to contractionary fiscal policy increases taxes and cuts spending.

Point 8) Page 3: My remark refers to the four components of the fiscal policy. The first two are clear but the rest requires more explanations. Fiscal policy refers to above all taxes and state expenses and to managing public assets to gain control and to influence the division of income as well as the general level of economy activity of a given country. FDI activity is treated rather as the part of widely understood economic policy and not as strict fiscal policy. A similar remark concerns the component of Debt/Surplus Management. It should be rethought if this is a component of this policy or rather it is the result of it.

Response for Point 8): The correction is correct; there should only be two components of fiscal policy, as stated by reviewer 2. As for the following two points, the government budget is the essential instrument for government spending policies. The budget is also used to finance the deficit, meaning that: Government budgets are the most critical instruments that embody government spending policies (Rodríguez-Pose; 2020). The budget is also used to finance the deficit to fill the gap between government spending and revenue; (i). Investment and policies: The optimal level of domestic and foreign investment is needed to maintain economic growth. In recent years, international capital flows or Foreign Direct Investment (FDI) has increased dramatically and has become a tool to integrate the domestic economy with the global economy (Watson. et all, 2014); (ii). Debt / Surplus Management: If the government receives more than what is spent, it is called a surplus. However, if the government spends more than income, then that is called a deficit. To finance the deficit, the government must borrow from domestic or foreign sources, or another option that can be taken is to print money to fund the deficit (Rodríguez & Garcilazo; 2015).

Point 9) Page 4: In the theoretical part the author presents the empirical examples concerning Indonesia. In my opinion the theoretical part of the article should be clearly divided from the empirical one.

Response for Point 9): Good advice and I'll remove that theoretical explanation.

Point 10) Page 4: In the references (Thornton, 2001) please reconsider the style of this sentence. Remember that Patricia is the name similarly to Malcolm and not a surname.

Response for Point 10):  I changed it to Patricia H. Thornton (2001) “Personal versus Market Logics of Control: A Historically Contingent Theory of the Risk of Acquisition.” Organization Science 12, no. 3: 294–311.10.1287/orsc.12.3.294.10100

Point 11).Taking into consideration the methodical approach and assumptions concerning the analysis of three ASEAN countries (Indonesia, Vietnam, Cambodia) I am of the opinion that there should be references and data referring to all of them. It concerns: table 1, fig 2, 3,4, table 2.

Response for Point 11): I added according to his suggestion which is:

Table 1. Source: Law Number . 40 the YEAR 2007. (2007-). About The Limited Company Ministry of Law and Human Rights of the Republic of Indonesia. In Article 8 paragraph (2) letter a, it is necessary to clarify the nationality of the founders to establish a company. An Indonesian legal entity in a company is established by an Indonesian citizen or an Indonesian legal entity. However, foreign nationals or foreign legal entities are allowed to establish an Indonesian legal entity in the form of a Company as long as the law governing the Company's line of business allows, or the establishment of the Company is regulated by a separate law. If the founder is a foreign legal entity, the number and date of legalization of the founding legal entity is a document similar to that, including a certificate of incorporation. If the founder is a state or regional legal entity, a government regulation concerning participation in the Company or a provincial law concerning regional involvement in the Company is required.

Figure 2. Indonesia's Economic Growth; Source: World Bank 2020

The World Bank that the direction of Indonesia's economic growth will continue to decline to the level of 4.9% in 2020, due to 1) Low productivity; 2) The slowdown in the growth of the force work; 3) Decrease in commodity prices; 4) Continuation of the US and China Trade War. China's economy fell by 1%, then Indonesia's GDP can fall by 0.3%. In the case of the 2009 global recession, global economic growth down by 6.2% from 2007, commodity prices fell, and the Indonesian economy slowed by 1.7%. Stimulus It is believed that monetary and fiscal conditions in Indonesia will be limited in the future. Projecting the opportunity for a 7DRR decrease is relatively limited, which is only a 25 to 5% level in 1H20F. In assessing that the positive effects of lowering interest rates are fading and even aggressive declines can trigger capital outflows in the future. Fiscal stimulus from the plan to reduce corporate tax from 25% to 20% is estimated to be effectively implemented in 2021. Therefore, lowering the projected corporate earnings growth from 8% - 10% in FY19E to 5.28% - 7.98% in FY19E FY20F. In my view, investment risk in Indonesia still lies in the : 1) The Current Account Deficit (CAD) is still widening; 2) The weakening of the exchange rate, which is still estimated to the level of 15,000 in FY20F; 3) Capital Outflow. Every year, Indonesia at least still needs external funding of around USD 16 billion to finance the CAD, where this value will increase in an outflow. On the other hand, the low level of labor productivity and the inefficiency of the bureaucracy in Indonesia have pushed down the contribution of FDI to Indonesia's GDP since 2014. The positive impact of the trade war is more beneficial for Vietnam, Thailand, Malaysia, Singapore, Taiwan. 23 Chinese companies that went public have announced plans for Vietnam expansion, the remaining 10 to Cambodia, India, Malaysia, Mexico, Serbia, and Thailand in 2019. Researchers believe that FDI growth is one of the essential indicators to increase investor confidence shares amid the very lack of catalysts for the Indonesian stock market at this time.

Figure 3. Effectiveness of Monetary policy in Indonesia

Source: Report of the Central Bank of Indonesia and the Central Statistics Agency 2019

Figure 4. : Indonesian imports and exports. Source: National Central Agency Report. 2019.

Table 2. References and sources have been submitted.

Point 12). The remarks concern the titles of tables and figures:

Response for Point 12): - table 1: there is no information which year(s) this data concerns.

I have provided the information according to the suggestions and input in point 11,

- figure 2: Projected decline in GDP 2010-2022? The data concerning the year 2010 cannot be projected because they are a fact. Now we live in 2021. You must give information in the title of this figure what country the data refers to.

Agree with the suggestions and inputs, so I changed it to Indonesia's Economic Growth; Source: World Bank 2020. Here I also emphasize and assess economic growth according to the advice in point 11.

- Figure 3: You cannot title this figure: "efficiency of monetary policy". There is only a presentation on current account balance not the whole efficiency or the lack of it. Please add the information about what countries and year(s)) it refers to.

I add, according to his suggestion, as follows: Figure 3. Effectiveness of Monetary policy in Indonesia. Source: Report of the Central Bank of Indonesia and the Central Statistics Agency 2019

- Figure 4 is titled "Indonesian imports and exports". Please add the years and the name of currency.

Figure 4: Indonesian imports and exports. Source: National Central Agency Report. 2019. The currency exchange rate is the US Dollar.

- Number of figures 5 is doubled,

Fixed.

- There is no proper name of fig 5, page 10: Indicators? Specify the name of the indicator and give the information about the units.

Figure 5 is used as figure 6: World Banks and World Development Indicators Foreign Direct Investment (FDI) In Southeast Asia (ASEAN) 2004-2018.

Foreign Direct Investment (FDI) Net in Flows Percent of GDP, 5 Year Moving Average

- The title of table 2 is not proper. It ought to be: Top exports of goods in Vietnam in…. years.

Agree on the suggestions for improvement: Top exports of goods in Vietnam (USD) Millions 2019.

- Fig 6: Foreign direct investment in Vietnam in the years ….. [ units].

Fig 6: Foreign Direct Investment in Vietnam in The Years 2014-2018

- The title of Fig 7: there is no information on the years this figure concerns. The values ​​of the horizontal lines are not given.

Figure 7 was changed to figure 8: Product Financial Assets to GDP in ASEAN 2019.

- Page 16 (one figure), 17 (two figures). There are no titles of these figures.

Figure: Simulation Game Theory Application and Government as A-State Institution Initiate and follow 0 to 1

Figure: Simulation Game Theory Application and Government as A-State Institution Implement and Not Implement 0.5 to 1

 Figure: Simulation Game Theory Application and Government as A-State Institution Initiate and follow -1 to 0 and 1

Point 13) The separated part of this article entitled” Conclusions and recommendations” ought to be prepared. So far the conclusion has been done, but please find some recommendations for Indonesia, Vietnam and Cambodia institutional policy and for ASEAN as an.

Response for Point 13): Here I provide additional recommendations in conclusion:

Recommendations that can be made based on the above topics are Institutional policy reforms through benchmarking to ASEAN countries such as Indonesia, Vietnam and Cambodia need to be carried out.

Evaluate and improve the ability to exist in human resources in specific sectors through the empowerment of Human Resources by the government. In the case of constitutional policy, there is needed to be independence and transparency to maintain policy reliability. In terms of monetary policy, it is necessary to determine the period in terms of increasing the benchmark interest rate because it can cause a decrease in GDP if done repeatedly. In the case of fiscal policy, a comprehensive budgetary plan can be carried out, namely by reducing taxes that aim to stimulate economic growth and attract investment.

Point 14) Please indicate the main goal of this article, the research questions or hypotheses and try to refer to them at the end of the article.

Response for Point 14): The purpose of this article is to explore in-depth how the institutions that ASEAN-Indonesia, Vietnam, and Cambodia have built in the face of the trade war between China and the USA to obtain input and policy recommendations. This article sees that ASEAN-Indonesia, Vietnam, and Cambodia still have coordination line problems institutions, Human Resources issues, budget issues, institutional arrangements, and ideal division of tasks that can optimize the direction of the country's policies. This article will provide significant benefits for the countries that are the sample of this research as a reflection to improve institutions, both for the ASEAN Community and other issues.

Best,

Fransiskus X Lara Aba 

Round 2

Reviewer 2 Report

Please, try to use the passive voice in scientific texts. In your work one can find: "While additional information about the trade war that I added in the introductory chapter, namely:..." 

Author Response

Dear reviewer 2, 

Thank you very much for  your correction and suggestion. 

Here I am correcting it by using passive sentences in "While additional information about the trade war that I added in the introductory chapter," as follows: 

Reviewer 2 Comments Point:

Comments and Suggestions for Authors

Please, try to use the passive voice in scientific texts. In your work, one can find: "While additional information about the trade war that I added in the introductory chapter, namely:..."

Response Author : Thank you very much for the corrections and suggestions. Here I am correcting it by using passive sentences in "While additional information about the trade war that I added in the introductory chapter," as follows:

While additional information about the trade war that I added in the introductory chapter, namely: The trade war is a term used by media and repeated by politicians and economists for United States (US) action who have raised import duties imports of manufactured products China and some other countries and got a backlash from the destination country. Action against China is taken because the US considers the government has jeopardized the interests of US nationals. China's export volume to the United States far exceeds the proportion of imports, so the US trade deficit with China since 2010, increasingly increased. To reduce the deficit trade, since June 2018 US government has launched a strategy to increase import duties, especially towards China. In early May 2019 US Government increased tariffs on imports of Chinese commodities worth 200 billion US dollars from 10% to 25%. 

China responds by imposing new import tariffs as of June 1, 2019, for items belonging to the US 60 billion US dollars. China, too, is considering stopping purchasing the production of agriculture and Boeing aircraft from The US. This trade war entered a new chapter when Google from the US decided android system connection with Chinese mobile phone manufacturers, Huawei, to government prohibition. Ban this has an impact significant for product users Huawei. Hundreds of millions of Huawei phones can't anymore access exclusive services Google-like Gmail, Youtube, play store, and google maps. US mobile parts company, Lumentum has also decided to stop sending spare parts to Huawei. Intelligence The US believes Huawei is supported by the military, so that it is suspected of providing intelligence services for the Chinese government.

 Best,

Fransiskus Aba